

# Soft and hard skills identification: insights from IT job advertisements in the CIS region

Andrei Ternikov

HSE University, Moscow, Russia

## ABSTRACT

Labor market transformations significantly affect the sphere of information technologies (IT) introducing new instruments, architectures, and frameworks. Employers operate with new knowledge domains which demand specific competencies from workers including combinations of both technical ("hard") and non-technical ("soft") skills. The educational system is now required to provide the alumni with up-to-date skill sets covering the latest labor market trends. However, there is a big concern about the self-adaptation of educational programs for meeting the companies' needs. Accordingly, frequent changes in job position requirements call for the tool for in-time categorization of vacancies and skills extraction. This study aims to show the demand for skills in the IT sphere in the Commonwealth of Independent States (CIS) region and discover the mapping between required skill sets and job occupations. The proposed methodology for skills identification uses natural language processing, hierarchical clustering, and association mining techniques. The results reveal explicit information about the combinations of "soft" and "hard" skills required for different professional groups. These findings provide valuable insights for supporting educational organizations, human resource (HR) specialists, and state labor authorities in the renewal of existing knowledge about skill sets for IT professionals. In addition, the provided methodology for labor market monitoring has a high potential to ensure effective matching of employees.

## INTRODUCTION

The labor market in the digital era provides many challenges for the educational system and potential employees. For example, companies adapt to modern technological changes and competing environments demanding a broader range of skills from their workers. Accordingly, new knowledge domains stimulate the creation of new job tasks which require the combination of competencies from different occupations. Hence, the educational curricula should be flexible for these transformations to provide alumni with up-to-date skills.

The sphere of information technologies (IT) is especially involved in the processes mentioned above. IT professionals are required to possess particular combination of skills.

Corresponding author
Andrei Ternikov, aternikov@hse.ru

Therefore, recruiters demand not only technical ("hard") but also non-technical ("soft") skills which are significant for an IT career (*Litecky, Arnett & Prabhakar, 2004*; *Johnson, 2016*; *Kappelman et al., 2016*; *Matturro, Raschetti & Fontán, 2019*; *Dubey & Tiwari, 2020*). The last includes, for example, communication abilities, leadership, teamwork, time management. Thus, IT industry representatives and human resource (HR) specialists introduce sets of competencies providing information in job advertisements in order to hire the most suitable specialists.

A wide range of research examines the IT sphere due to its feasibility to technologies provided and the simplicity of skills generalization. Academic literature accumulates knowledge about the demand side of the IT labor market and investigates issues connected with skills identification from online job advertisements (*Papoutsoglou et al., 2019*; *Khaouja, Kassou & Ghogho, 2021*). Firstly, related research highlights the importance of online job advertisement platforms in the process of labor market monitoring. Secondly, job advertisements data are used for clustering vacancies among job occupations (*De Mauro et al., 2018*; *Gurcan & Cagiltay, 2019*; *Pejic-Bach et al., 2020*). Unstructured textual fields from job titles and their extended descriptions help to extract specific topics with keywords and run algorithms for vacancies' clusterization. Accordingly, text mining techniques are widely applied for the identification of core competencies. Thirdly, the data obtained from hiring platforms are matched with official classifiers for occupations and skills and then unified with labor market representation (*Amato et al., 2015*; *Botov et al., 2019*; *Boselli et al., 2018*; *Colombo, Mercorio & Mezzanzanica, 2019*; *Lovaglio et al., 2018*). Existing studies use several open-source databases with processed "soft" and "hard" skills for the mentioned tasks. Authors point out that non-technical skills are highly required along with the technical ones but their interrelation is under-investigated (*Verma et al., 2019*; *Radovilsky et al., 2018*; *Fareri et al., 2020*; *Pejic-Bach et al., 2020*; *Litecky et al., 2009*; *Poonnawat, Pacharawongsakda & Henchareonlert, 2017*).

IT industry expects to hire candidates who operate with broad knowledge domains. Nevertheless, questions about the need for "hard" and "soft" skills are raised separately in related works. Existing studies investigate such skills without finding the interrelation between technical and non-technical competencies. Moreover, the variety of techniques and methodological pipelines allows generalizing existing approaches in terms of knowledge extraction, standardization, and skill set identification. The main problem is still to investigate which combinations of "soft" and "hard" skills are demanded by the companies hiring IT professionals. In this regard, the paper systematizes existing skill identification pipelines. In addition, it presents an approach which allows to find and visualize the interrelation between non-technical skills and combinations of technical competencies characterizing particular IT occupations using the data from the Commonwealth of Independent States (CIS) labor market.

This research aims to identify "soft" and "hard" skills domains for IT occupations. Therefore, this study focuses on the following research questions:
1. Which "soft" and "hard" skills are in-demand for IT specialties in the CIS region?
2. Which aggregated job families can be identified based on required competencies?

3. Which in-demand combinations of technical and non-technical skills can be highlighted?

The paper fills a gap in the semi-automatic analysis of competencies in the labor market based on online job advertisements and provides an extended skill-based approach to define clusters of vacancies and combinations of skills in them. I unify "soft" and "hard" skills in IT specialties, aggregate job families in attribution to required competencies, and investigate which combinations of them are needed in IT job occupations.

The significant result of the paper relates to the proposition of non-technical skills that are associated with several combinations of technical skills by job occupations. Obtained information has a high potential to be implemented in the educational process and the maintenance of already existing learning standards. Also, a particular interest of educational institutions concerns to support of their alumni in the process of hiring. According to the results of this study, the possession of only technical skills among IT professions cannot guarantee successful job matching. Thus, the competitiveness of the potential job candidates is compensated by the presence of well-matched technical and non-technical skills combinations. This paper is a continuation of our prior research (*Ternikov & Aleksandrova, 2020*). The main improvements from that study imply the following aspects: the broader data-set is used; the implementation of semi-automatic procedures for skills standardization for synonyms and generalized terms detection; job occupations identification is based on skills clustering and association mining; the analysis of non-technical and technical skills including their interrelation.

The contribution of this paper is five fold. First, I present an in-depth review of methodological pipelines for skills extraction and identification from IT job postings. Second, I provide the skill-driven pipeline for job advertisement analysis that allows to find the interrelation between skills for particular job occupations. Third, I use experimental setting to analyse the data from the CIS region that is poorly investigated. Fourth, I identify the combinations of demanded "soft" and "hard" skills for IT professionals using association mining techniques. I provide a common understanding of labor market requirements in a measurable form including not only the frequency of the presented competencies but also the probability of the co-occurrence of their combinations. Fifth, I present insights in terms of skills identification and "soft" skills' importance for future research directions.

The rest of the paper is structured as follows. Related works section provides an overview of studies on skill identification. The methodology, research pipeline, data collection, and data processing are described in the following section. The last three sections contain experiment results, discussion and conclusions.

## RELATED WORK

Existing studies extract knowledge from IT job advertisements using different methodological pipelines. Researchers identify and aggregate skills into sets related to job occupations and introduce several experiments. In particular, competence frequency and skill sets interrelation are analyzed. Moreover, some papers provide matching between

job occupations and competencies. Generally, the process of skills identification from job advertisements consists of three phases: text processing, skill base mapping, and gathering skill sets related to job occupations. Accordingly, the choice of skills identification and aggregation approaches depends on the sample size, type of skills, and research focus. In this section, a review of recent studies in terms of methodological pipelines is provided. Related works are summarized in Table 1. The table is structured as follows: the first four columns indicate the groups of related works in attribution to the context of skills extraction and standardization (the sample size, approach for skills extraction, and skill types used in the analysis). The following two columns aggregate the related works connected to methods of skill sets and job occupations detection. The last three columns point out the main focus of the papers from the previous two columns.

## Skills extraction and standardization

Related research highlights four main approaches for extraction and standardization of skills from job advertisements depending on the degree of human involvement, namely, content analysis, frequency count, topic modeling, and classification. A detailed description of each approach applied to the IT sector is provided below.

### Content analysis

Content analysis is based on a semi-manual aggregation of qualitative data. It allows to build concepts and assign related topics to each data entry (job advertisement description). Commonly, it consists of two stages. The first stage of the procedure is frequent terms extraction, the second stage—manual mark-up and pre-defined topics (concepts) allocation. In general, the sample size for this method does not exceed 1,000 data entries due to a limited number of job postings obtained for the research.

In the context of skills standardization, the content analysis provides more precise results comparing with other automatic algorithms (*Cegielski & Jones-Farmer, 2016*). The advantage is the correct classification of skills with different notations and the same meaning, and vice versa. However, this method is time-consuming and not applicable to large datasets. Accordingly, related research, which applies content analysis, investigates only the frequency count of extracted skills (*Hussain, Clear & MacDonell, 2017*; *Chang, Wang & Hawamdeh, 2019*; *Matturro, 2013*; *Chaibate et al., 2019*; *Steinmann, Voigt & Schaeffler, 2013*; *Sodhi & Son, 2010*). Thus, the issues which cover interrelation between skills and their mapping with job occupations are under-examined.

### Frequency count

Frequency count and manual processing of textual data are generally used as a part of a methodological pipeline. However, several studies provide this approach separately on the stage of skills standardization. I can identify two main directions for its usage.

Firstly, skills extraction and ambiguous terms reduction (*Ahmed, Capretz & Campbell, 2012*; *Gardiner et al., 2018*; *Daneva, Wang & Hoener, 2017*; *Bensberg, Buscher & Czarnecki, 2019*; *Ternikov & Aleksandrova, 2020*). Researchers use standard text pre-processing techniques such as punctuation removal, letters lowercase, tokenization. Then, authors

**Table 1  Literature overview of pipelines for IT job advertisements analysis.**

| Skills extraction and standardization | | Type of skills | | Skill sets/occupations detection (References) | | | Results focus | |
|---|---|---|---|---|---|---|---|---|
| Sample size | Approach | Hard | Soft | Skill base mapping | Clustering | Skill frequency | Co-occurrence | Matching |
| $<10^3$ | Content Analysis | + | + | *Cegielski & Jones-Farmer (2016); Hussain, Clear & MacDonell (2017); Chang, Wang & Hawamdeh (2019); Sodhi & Son (2010)* | — | *Hussain, Clear & MacDonell (2017); Sodhi & Son (2010); Chang, Wang & Hawamdeh (2019)* | — | — |
| | | — | + | *Matturro (2013); Chaibate et al. (2019); Steinmann, Voigt & Schaeffler (2013)* | — | *Matturro (2013); Chaibate et al. (2019); Steinmann, Voigt & Schaeffler (2013)* | — | — |
| $10^2$–$10^5$ | Frequency Count | + | + | *Gardiner et al. (2018); Skhvediani et al. (2021); Wowczko (2015); Fareri et al. (2020); Verma et al. (2019); Ternikov & Aleksandrova (2020); Papoutsoglou, Mittas & Angelis (2017); Hiranrat & Harncharnchai (2018); Bensberg, Buscher & Czarnecki (2019); Brooks, Greer & Morris (2018)* | — | *Verma et al. (2019); Wowczko (2015); Gardiner et al. (2018); Bensberg, Buscher & Czarnecki (2019); Papoutsoglou, Mittas & Angelis (2017); Hiranrat & Harncharnchai (2018); Skhvediani et al. (2021); Brooks, Greer & Morris (2018)* | *Papoutsoglou, Mittas & Angelis (2017)* | *Verma et al. (2019); Brooks, Greer & Morris (2018); Fareri et al. (2020); Ternikov & Aleksandrova (2020)* |
| | | — | + | *Florea & Stray (2018); Daneva, Wang & Hoener (2017); Ahmed, Capretz & Campbell (2012)* | — | *Florea & Stray (2018); Daneva, Wang & Hoener (2017)* | — | *Ahmed, Capretz & Campbell (2012)* |
| $10^3$–$10^6$ | Topic Modeling | + | + | *Litecky, Igou & Aken (2012)* | *Litecky et al. (2009); Xu et al. (2018); Gurcan & Cagiltay (2019); Pejic-Bach et al. (2020); Radovilsky et al. (2018); Poonnawat, Pacharawongsakda & Henchareonlert (2017); Wu, Shi & Yang (2017)* | *Xu et al. (2018); Litecky, Igou & Aken (2012)* | *Wu, Shi & Yang (2017); Gurcan & Cagiltay (2019)* | *Litecky et al. (2009); Radovilsky et al. (2018); Poonnawat, Pacharawongsakda & Henchareonlert (2017); Pejic-Bach et al. (2020)* |
| | | + | — | — | *Debortoli, Müller & vom Brocke (2014); De Mauro et al. (2018)* | — | — | *Debortoli, Müller & vom Brocke (2014); De Mauro et al. (2018)* |
| $>10^4$ | Classification | + | + | *Giabelli et al. (2020); Colace et al. (2019); Colombo, Mercorio & Mezzanzanica (2019); Amato et al. (2015); Hoang et al. (2018); Botov et al. (2019); Cao et al. (2021); Boselli et al. (2018); Lovaglio et al. (2018); Karakatsanis et al. (2017)* | *Börner et al. (2018); Jia et al. (2018)* | *Jia et al. (2018); Cao et al. (2021); Giabelli et al. (2020)* | *Börner et al. (2018)* | – |
| | | — | + | *Sayfullina, Malmi & Kannala (2018)* | *Calanca et al. (2019)* | *Calanca et al. (2019)* | — | — |

manually extract or remove frequent items obtained after TF-iDF (term frequency—inverse document frequency) procedure.

Secondly, data entries categorization, annotation, and labeling (*Florea & Stray, 2018*; *De Mauro et al., 2018*; *Skhvediani et al., 2021*; *Brooks, Greer & Morris, 2018*; *Wowczko, 2015*; *Fareri et al., 2020*; *Papoutsoglou, Mittas & Angelis, 2017*; *Verma et al., 2019*). Authors tokenize textual data and use contiguous sequences of *n* words (*n*-grams). Then, researchers manually validate obtained items by comparing their frequencies and aggregating them in groups of skills with the same meaning (synonyms).

Summing up, manual processing of frequent terms allows to aggregate a larger amount of information comparing with content analysis. The processing is based on the parts of job descriptions which consist of several tokens of text sequences. Moreover, the procedure is independent of pre-defined concepts. The last-mentioned allows using computerized approaches for matching skills and job occupations (*Fareri et al., 2020*; *Brooks, Greer &*

*Morris, 2018*; *Ahmed, Capretz & Campbell, 2012*; *Papoutsoglou, Mittas & Angelis, 2017*; *Verma et al., 2019*; *Ternikov & Aleksandrova, 2020*).

### Topic modeling

Topic modeling is an automatic approach for allocating unstructured information into groups with common semantic patterns. In the context of skills extraction, this method operates with words (or sequences of words) and their occurrence in job advertisements. Existing studies on topic modeling provide no manual correction of obtained terms comparing with already mentioned approaches. However, the description of the resulting topics is carried with manual validation by domain experts.

Topic modeling is flexible and covers different research objectives. For example, some authors analyze the popularity of job skills (*Litecky, Igou & Aken, 2012*; *Xu et al., 2018*). Other authors discover competencies interrelation (*Wu, Shi & Yang, 2017*; *Gurcan & Cagiltay, 2019*). Moreover, the use of unsupervised models for topic modeling helps to match skill sets and job occupations (*Pejic-Bach et al., 2020*; *Radovilsky et al., 2018*; *Debortoli, Müller & vom Brocke, 2014*; *De Mauro et al., 2018*; *Litecky et al., 2009*; *Poonnawat, Pacharawongsakda & Henchareonlert, 2017*).

### Classification

Classification tasks for skills extraction are raised in more recent research. Authors use supervised machine learning algorithms in order to match particular job descriptions with already labeled databases of competencies. This approach is mainly used with large datasets. Accordingly, the portion of data is marked-up manually, and then, the results are validated.

The initial task of these works is to match job advertisement descriptions with a standardized database of skills or occupations. Hence, the algorithmic pipeline depends on a domain basis for skills validation. Some authors use knowledge graphs indicating the co-occurrence of raw terms (*Jia et al., 2018*; *Giabelli et al., 2020*; *Börner et al., 2018*; *Colace et al., 2019*; *Boselli et al., 2018*). The others apply descriptions of skills from national occupational classifiers (*Cao et al., 2021*; *Colombo, Mercorio & Mezzanzanica, 2019*; *Karakatsanis et al., 2017*; *Lovaglio et al., 2018*; *Botov et al., 2019*; *Amato et al., 2015*). The rest—use manual mark-up for extracted knowledge domains (*Calanca et al., 2019*; *Sayfullina, Malmi & Kannala, 2018*; *Hoang et al., 2018*).

## Soft and hard skills detection

The analysis of the required skills obtained from online job advertisements is widely investigated in the research literature. The authors highlight two types of skills: "soft" and "hard". These skills are analyzed together or separately depending on research objectives.

Some authors use only one type of mentioned competencies and do not provide interaction between skills of different types. Specifically, related works which use only "soft" skills in the IT sector are generally mapping studies. The main objectives follow the skill base dictionary creation and skill frequency count (*Sayfullina, Malmi & Kannala, 2018*; *Calanca et al., 2019*; *Daneva, Wang & Hoener, 2017*; *Florea & Stray, 2018*; *Ahmed, Capretz & Campbell, 2012*; *Matturro, 2013*; *Chaibate et al., 2019*; *Steinmann, Voigt & Schaeffler,*

*2013*). In contrast, studies operating only with "hard" skills focus on the characteristics of skill set related job families (*Debortoli, Müller & vom Brocke, 2014*; *De Mauro et al., 2018*).

The other existing studies use both "soft" and "hard" skills. Some authors analyze mixed skill sets but without distinction between their types (*Colace et al., 2019*; *Karakatsanis et al., 2017*; *Lovaglio et al., 2018*; *Botov et al., 2019*; *Boselli et al., 2018*; *Amato et al., 2015*; *Jia et al., 2018*; *Börner et al., 2018*; *Pejic-Bach et al., 2020*; *Litecky et al., 2009*; *Poonnawat, Pacharawongsakda & Henchareonlert, 2017*; *Xu et al., 2018*; *Wowczko, 2015*; *Fareri et al., 2020*; *Bensberg, Buscher & Czarnecki, 2019*; *Ternikov & Aleksandrova, 2020*; *Verma et al., 2019*). Another group of papers provides separation in two ways. Firstly, authors use manual mark-up and ready skill bases before the stage of skill sets detection (*Cao et al., 2021*; *Hoang et al., 2018*; *Skhvediani et al., 2021*; *Papoutsoglou, Mittas & Angelis, 2017*; *Hiranrat & Harncharnchai, 2018*; *Cegielski & Jones-Farmer, 2016*). Secondly, researchers obtain "soft" skills mapping using domain experts correction after classifying or clustering job advertisements (*Colombo, Mercorio & Mezzanzanica, 2019*; *Litecky, Igou & Aken, 2012*; *Gardiner et al., 2018*; *Brooks, Greer & Morris, 2018*; *Hussain, Clear & MacDonell, 2017*; *Sodhi & Son, 2010*; *Gurcan & Cagiltay, 2019*; *Radovilsky et al., 2018*; *Giabelli et al., 2020*; *Wu, Shi & Yang, 2017*).

## Identification of job occupations

Analysis of skills identification pipelines includes the stage of aggregation of job occupations (job related skill sets). Recent research could be separated into groups based on the approach of data processing. The first group uses skill base mapping which helps to match information from job postings with descriptions obtained from official classifiers of occupations and skills. The second group includes clustering methods which allow merging similar vacancies containing the same required skill sets.

### *Skill base mapping*

The presence of official classifiers allows to distribute the raw data from job postings among groups created by domain experts. Some authors use job titles and their descriptions from vacancies in order to match them with ESCO (European Skills/Competences, qualifications and Occupations), ISCO (International Standard Classification of Occupations), or O*NET (Occupational Information Network) classifiers (*Boselli et al., 2018*; *Cao et al., 2021*; *Colombo, Mercorio & Mezzanzanica, 2019*; *Lovaglio et al., 2018*; *Colace et al., 2019*; *Papoutsoglou, Mittas & Angelis, 2017*; *Giabelli et al., 2020*). Others use the structure of job advertisements provided on hiring platforms and validate it with domain experts mark-up and professional standards (*Chang, Wang & Hawamdeh, 2019*; *Ternikov & Aleksandrova, 2020*; *Hiranrat & Harncharnchai, 2018*; *Amato et al., 2015*; *Sayfullina, Malmi & Kannala, 2018*; *Litecky, Igou & Aken, 2012*; *Hoang et al., 2018*; *Botov et al., 2019*; *Brooks, Greer & Morris, 2018*). This research employs text mining techniques such as TF-iDF and *n*-grams that are used for specific words and phrases extraction. By implementing algorithms of classification, the authors highlight several groups of IT occupations where the most frequent technical and non-technical skills are outlined. Such division of skills is obtained from a given classifier and manual processing.

The other group of related works uses manually obtained expert-based keywords and content analysis or classifying vacancies and identifying the most frequent skills (*Bensberg, Buscher & Czarnecki, 2019*; *Fareri et al., 2020*; *Gardiner et al., 2018*; *Ahmed, Capretz & Campbell, 2012*; *Florea & Stray, 2018*; *Cegielski & Jones-Farmer, 2016*; *Hussain, Clear & MacDonell, 2017*; *Matturro, 2013*; *Daneva, Wang & Hoener, 2017*; *Chaibate et al., 2019*; *Steinmann, Voigt & Schaeffler, 2013*; *Sodhi & Son, 2010*; *Verma et al., 2019*; *Skhvediani et al., 2021*; *Wowczko, 2015*). The authors examine digital-oriented vacancies and structure competencies on a qualitative basis. For example, online vacancies are chosen and filtered by job position names. After implementing text mining analysis, the key skills are grouped into several categories with the use of official classifiers.

### Clustering

The structure of online job postings does not resemble official classifiers in every case. Several authors obtain their categorizations of vacancies based on clustering and data-driven approaches. However, the choice of an appropriate clustering algorithm depends on the data structure and information available. For example, researchers use Latent Dirichlet Allocation (*De Mauro et al., 2018*; *Gurcan & Cagiltay, 2019*) and Hierarchical Clustering (*Pejic-Bach et al., 2020*; *Litecky et al., 2009*).

The process of key skills identification in most cases is determined by keywords extraction (particular skill set). In addition, the number of clusters is set experimentally and specific groups of competencies are corrected manually. Commonly, the number of groups varies from eight to 10. Accordingly, two directions of the research can be outlined. The first one includes the creation of groups of vacancies based on words obtained from job descriptions (*De Mauro et al., 2018*; *Pejic-Bach et al., 2020*; *Litecky et al., 2009*; *Xu et al., 2018*; *Börner et al., 2018*). For instance, some keywords are combined into skill sets and redistributed between clusters. The second direction uses skill sets representation as vectors of words (*Gurcan & Cagiltay, 2019*; *Debortoli, Müller & vom Brocke, 2014*; *Wu, Shi & Yang, 2017*; *Radovilsky et al., 2018*; *Poonnawat, Pacharawongsakda & Henchareonlert, 2017*; *Calanca et al., 2019*; *Jia et al., 2018*). However, authors create groups of skills based on a clustering algorithm for particular job occupations.

### Application of skills identification and research gap

Most related works gather knowledge about the relative frequency of in-demand skills. Precisely, combining quantitative and qualitative approaches, authors identify existing competencies on the labor market in the IT sphere. Moreover, existing studies provide listings of technical and non-technical competencies analyzing their combinations. However, the co-occurrence is provided on the basis of mixed-type skill sets or among technical competencies (*Papoutsoglou, Mittas & Angelis, 2017*; *Wu, Shi & Yang, 2017*; *Börner et al., 2018*; *Gurcan & Cagiltay, 2019*). The other studies implement matching experiments between obtained skill sets and the professional structure of the job market. Authors point out that non-technical skills are highly required along with the technical ones but do not investigate their interrelation (*Verma et al., 2019*; *Fareri et al., 2020*; *Radovilsky et al., 2018*; *Pejic-Bach et al., 2020*; *Litecky et al., 2009*; *Poonnawat, Pacharawongsakda & Henchareonlert, 2017*).

Despite the abundance of studies in this research field, interactions and combinations between technical and non-technical skills inside different job profiles are under-examined (*Ahmed, Capretz & Campbell, 2012*; *Cao et al., 2021*; *Florea & Stray, 2018*; *Papoutsoglou, Mittas & Angelis, 2017*). Moreover, this paper goes beyond the vast majority of studies as it provides combinations of technical and non-technical skills based on classification and matching of job postings using association mining techniques.

## DATA AND METHODOLOGY

The research uses a semi-automatic methodology for in-demand skills identification among IT job occupations. The model pipeline of this study includes the stages of natural language processing, clustering analysis, and association mining (Fig. 1). The analysis was implemented in three main steps. Firstly, data were collected and processed for standardized skills extraction. Secondly, skill-based clustering analysis for job occupations detection was conducted using the hierarchical clustering algorithm. Thirdly, association mining for key combinations of "soft" and "hard" skills was carried out inside the obtained groups of job occupations. Each stage of the methodology is described in detail in the subsequent sections.

### Data collection and processing

The data were collected from one of the largest hiring platforms in the CIS (Commonwealth of Independent States) region named HeadHunter (https://www.hh.ru). Typical structure of online job advertisement (vacancy) includes the following main fields: vacancy ID, job name, specialization codes (from one to six professional area codes), publishing date, area (region), description (unstructured text), skills (the sets of 0 to 30 elements which consist of unstructured texts each up to 100 symbols). This study used the sample of IT vacancies with proposed skills (HeadHunter specialization "IT, Internet, Telecom") covering the 2015–2019 period, which contains 351,623 observations. In addition, the set of 3,034 unique frequent skills was extracted and prepared for further standardization.

The general logic of skill standardization follows the steps of similar terms finding (synonyms) and generalized terms aggregation (the common term for the particular subset of skills) (*Hoang et al., 2018*; *Hiranrat & Harncharnchai, 2018*; *Ternikov & Aleksandrova, 2020*; *Lovaglio et al., 2018*; *Gardiner et al., 2018*; *Brooks, Greer & Morris, 2018*; *Karakatsanis et al., 2017*; *Verma et al., 2019*). Moreover, the steps of matching abbreviations and multi-lingual terms processing are highlighted separately. To minimize the manual processing the following procedures were implemented:

1. splitting the terms by punctuation into smaller ones;
2. removal of exuberant punctuation and digits;
3. tracking the terms with white-spaces (words reordering and stemming);
4. finding of potential abbreviations by the first letters extraction from terms with white-spaces;
5. translation of terms with Cyrillic letters using Yandex.Translate API [1] (from Cyrillic-based text to English);
6. tokenizing and extracting $n$-grams ($n \in \{2, 3, 4\}$);

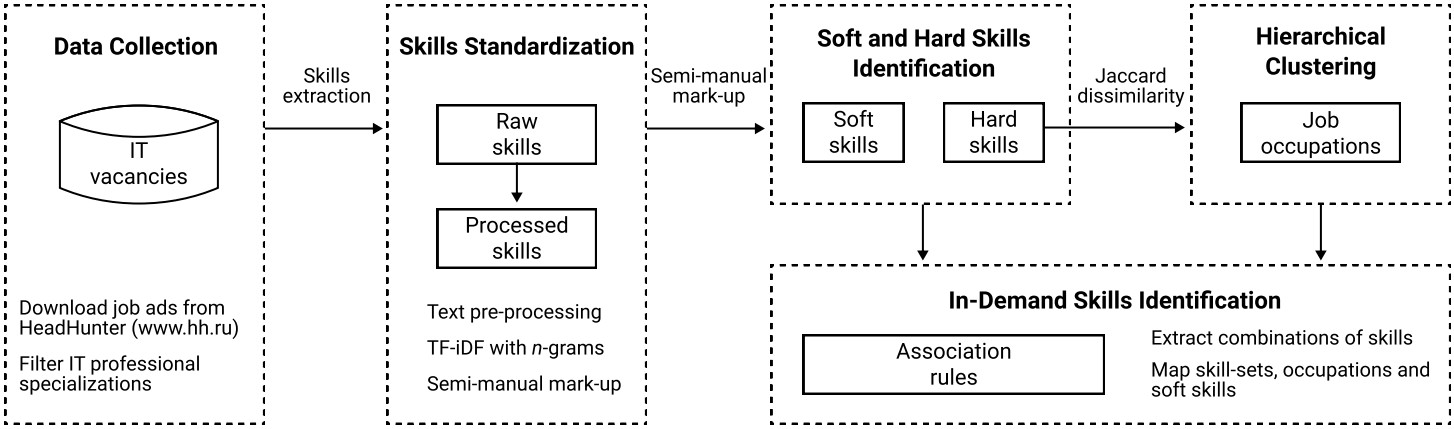

**Figure 1** **Research methodology pipeline.**

7. matching terms, tokens and *n*-grams based on TF-iDF (term frequency —inverse document frequency) approach;

8. manual correction and aggregation of terms into synonyms (different notation of the same skill) and generalized terms (common notation for different sub-skills).

At the endpoint of this stage 1,730 terms were obtained (including both synonyms and generalized terms) which were matched with the initial sample data. After the matching, 97.7% of vacancies remained with at least one matched term from the obtained dataset.

## Implementation of hierarchical clustering

In order to run a clustering algorithm based on the information about the competencies, the division between "soft" and "hard" skills was provided. The following analysis was based on the two-stage model which implies clusterization only over "hard" skills (*Litecky, Arnett & Prabhakar, 2004*).

From the initial sample of 1,730 terms, "soft" skills were semi-manually processed with the use of already introduced databases of non-technical skills (*Calanca et al., 2019*; *Sayfullina, Malmi & Kannala, 2018*). Accordingly, I have enriched the existing dictionaries of non-technical terms proposed in the other works within the HeadHunter specific notations of different skills. 94 "soft" skills were extracted and aggregated to 41 generalized groups. Then, the remaining "hard" skills were extracted and processed to identify only the competencies which were presented in the combinations with at least one of the processed "soft" skills. Thus, 544 technical skills were obtained.

Each skill is represented by a set of vacancy IDs. Jaccard indexes were calculated over all the paired combinations of "hard" skills (*Colace et al., 2019*; *Jia et al., 2018*; *Giabelli et al., 2020*). Jaccard index $J$ is the measure of similarity between two sets of objects $A$ and $B$ denoted as $J(A, B) = |A \cap B|/|A \cup B|$. To calculate these similarity measures within the reasonable computational time the MinHash procedure with 100 hash-functions was used (*Broder, 1997*). Based on the calculations, the dissimilarity square matrix was created

for clustering purposes. Each element of the matrix corresponds to the Jaccard distance $d_J(A, B) = 1 - J(A, B)$ between the sets of vacancy IDs for each skill.

Then, I used the Hierarchical Clustering procedure based on Weighted Pair Group Method with Arithmetic Mean (WPGMA) which is computationally easier for the detection of clusters with a different number of elements (*Pejic-Bach et al., 2020*). Formally, the distance $d$ between two combined clusters $i \cup j$, and the other cluster $k$ is defined as $d_{(i \cup j), k} = (d_{i,k} + d_{j,k})/2$. Thus, in order to obtain disconcordant clusters over "hard" skills only, WPGMA was implemented. The empirical choice of the number of clusters is justified in related research (*De Mauro et al., 2018*; *Debortoli, Müller & vom Brocke, 2014*; *Gurcan & Cagiltay, 2019*; *Pejic-Bach et al., 2020*; *Litecky et al., 2009*).

## Implementation of association mining

Association rules mining was proposed over the obtained clusters and terms. So, following the mentioned approach, suitable and more likely connected combinations of "hard" skills were matched with one appropriate "soft" skill. The assessment of the association rules for the sets $A$ (left-hand side) and $B$ (right-hand side) is based on the following indicators: $Support = P(A \cap B)$, $Confidence = P(A \cap B)/P(A)$, and $Lift = P(A \cap B)/(P(A) \cdot P(B))$. In this paper the following formal setting is used: left-hand side (lhs) relates to "hard" skills, right-hand side (rhs) implies one pre-processed "soft" skill (each rule interpretation is the chance of the match between the set of "hard" skills and the particular "soft" skill), "Confidence" threshold equals 0.001, "Support" threshold —0.0005. Finally, completed rules were merged with obtained clusters, and only rules where all "hard" skills are simultaneously presented in one cluster were analyzed.

# RESULTS

This section provides detailed results of each step of the methodological pipeline. Firstly, the most frequent "hard" and "soft" skills are identified. Secondly, clusters of professional skill sets are described. Thirdly, in-demand combinations of skills are listed and visualized by job occupations.

## Identification of soft and hard skills

The skills obtained after the data processing and standardization stage were separated into two groups: "hard" and "soft" skills. The formulations of technical competencies which have been translated into the English language were saved in lowercase. The relative frequencies over the whole range of IT vacancies are presented in the form of word clouds (Fig. 2).

## Clusters of job occupations

Hierarchical Clustering algorithm based on the WPGMA method was run over "hard" skills. As a result, 544 technical competencies were unequivocally distributed along 10 clusters. The number of clusters was chosen on the basis of internal validity scores for Hierarchical Clustering (Fig. 3) and in terms of preserving relatively the same number of representatives in each group of "hard" skills. Formally, the "Connectivity" metric should be minimized; "Dunn" and "Silhouette" —maximized.

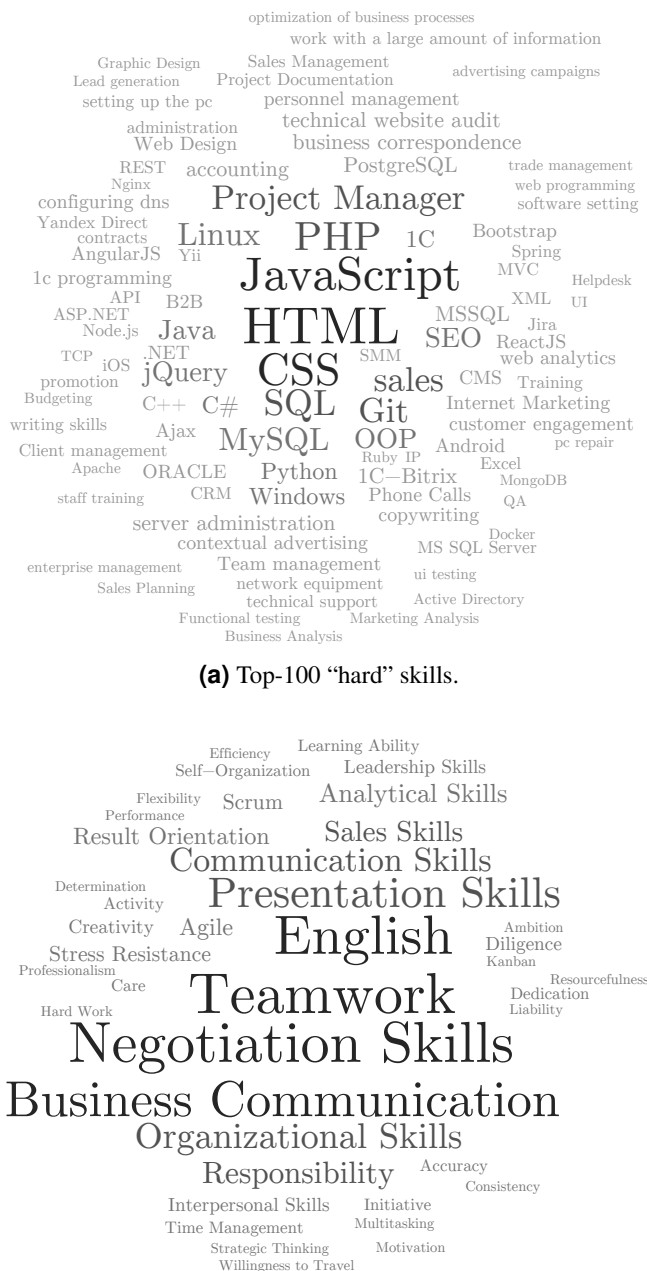

**(a)** Top-100 "hard" skills.

**(b)** "Soft" skills.

**Figure 2** **Word clouds of key skills in online job postings.**

The clusters themselves and the ten most frequent elements are presented in Table 2. The table includes the cluster name, the number of represented skills, and the 10 most frequent technical skills (according to their occurrence in analyzed vacancies). The names of clusters were introduced manually based on frequent terms and common ways of representation of IT groups in related works.

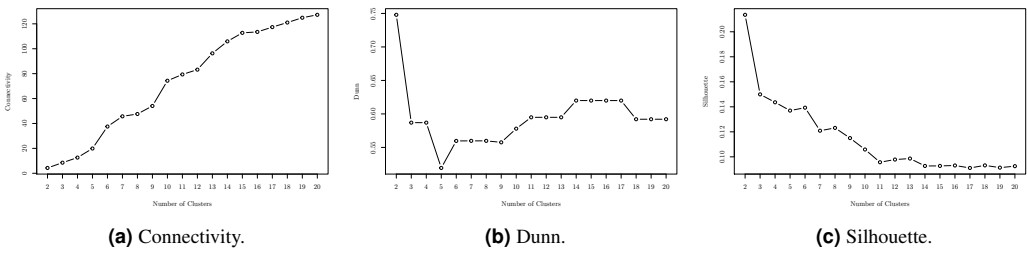

**(a)** Connectivity.          **(b)** Dunn.          **(c)** Silhouette.

**Figure 3  Internal validity scores for different number of clusters.** (A) Connectivity. (B) Dunn. (C) Silhouette.

**Table 2  Job occupation clusters.**

| # | Cluster name | N of skills | Top-10 "hard" skills |
|---|---|---|---|
| 1 | Analytics | 62 | 1C, accounting, 1c programming, trade management, optimization of business processes, enterprise management, Business Analysis, ERP, Reporting, create configuration 1c |
| 2 | Big Data & ML (#1) | 26 | Data Analysis, Machine Learning, Mathematical Statistics, Data Mining, statistical analysis, Mathematical Modeling, MATLAB, forecasting, Mathematical Programming, Mathematics |
| 3 | Big Data & ML (#2) | 73 | Linux, Java, Python, PostgreSQL, C++, Jira, Spring, Nginx, Apache, Ruby |
| 4 | Databases | 35 | SQL, C#, MSSQL, .NET, ORACLE, ASP.NET, MVC, MS SQL Server, SVN, MS Visual Studio |
| 5 | Engineering | 37 | Project Documentation, AutoCAD, System Analysis, assigning tasks to developers, gost, automation of processes, process control system, System Integration, Product Development, normative-technical documentation |
| 6 | Hardware | 64 | Windows, server administration, configuring dns, network equipment, setting up the pc, technical support, software setting, IP, TCP, pc repair |
| 7 | SEO | 60 | SEO, technical website audit, contextual advertising, Internet Marketing, copywriting, web analytics, Yandex Direct, SMM, administration, promotion |
| 8 | Support | 65 | sales, Project Manager, business correspondence, B2B, Phone Calls, personnel management, customer engagement, Team management, Client management, CRM |
| 9 | Testing | 81 | HTML, JavaScript, CSS, PHP, Git, MySQL, OOP, jQuery, 1C-Bitrix, Ajax |
| 10 | WebDev | 41 | Web Design, writing skills, UI, Graphic Design, UX, layout, CorelDRAW, editing, translation, rewriting |

Following the way of clusters aggregation in related research, the second cluster and the third one were merged due to the inclusion of close skills for "Big Data" and "Machine Learning" ("ML") respectively ("Big Data & ML (#1)" and "Big Data & ML (#2)") (*De Mauro et al., 2018*; *Gurcan & Cagiltay, 2019*; *Pejic-Bach et al., 2020*). The resulting list of 9 clusters relates to professional skill sets containing several specific skills required

by employers in the IT sector. The names of the clusters correspond to particular job occupations.

## Identification of in-demand skills for IT

The final stage of the analysis includes the aggregation of the obtained results after clustering and association mining steps. Identification of the most interdependent skill sets demands both understanding of common requirements for "hard" skills and finding an appropriate "soft" skill that has a high chance to be combined with the technical competencies.

Firstly, "soft" skills are left out. Then, the most required combinations (pairs) of "hard" skills are extracted using the Jaccard similarity matrix obtained before the clustering stage. Table 3 presents Top-5 most coherent combinations for each job occupation. The table indicates the cluster name in the first column, pairs of skills (the second and the third columns), and the Jaccard similarity for the proposed pair based on sets of vacancies where such skills are introduced.

Secondly, preserving relations between "soft" and "hard" skills poses a question regarding the most demanded non-technical competencies for different job occupations. In order to answer the stated question, the results of association mining analysis were used.

At first, only the rules containing "Lift" which exceeds unity were preserved. It allowed to maintain only relations of two skill sets with high chances to be required together. Then, all the rules were averaged by "Confidence" in attribution to different clusters to obtain an interpretation of the relative occurrence frequency of "soft" skill among combinations of "hard" skills connected to it. As a result, only 22 "soft" skills remained for 9 groups of job occupations. Visually the grid with average "Confidences" (in %) of obtained association rules is presented in Fig. 4. Each circle indicates the relative frequency of the particular "soft" skill among the sets of "hard" skills related to it in a specified job group. The size, the value, and the color saturation of each circle are all attributed to average "Confidences". In addition, Table 4 provides an extended description of the most probable and suitable combinations (by maximum "Confidence") of technical competencies for already extracted "soft" skills. The table lists the obtained rules indicating the "soft" skill (rhs), name of the cluster, "hard" skills (lhs), and measures of "Confidence", "Lift", and the number of vacancies where the rule is mentioned (N).

The grid provides some valuable insights into non-technical competencies that are highly required by IT companies in the labor market. For example, the most represented non-technical skills are "Agile", "Scrum", and "Organizational Skills". However, not all of them are required to the same degree for different professional groups. The most frequent technical skills relate to scripting and mark-up programming languages that are mostly used in web development. The other frequent skills cover the sphere of database administration and competencies related to project management.

According to non-technical skills, knowledge of the English language and teamwork skills are highly required. Namely, for SEO specialists "Communication Skills" are important, but for Web Developers ("WebDev") —"English" and "Creativity". At the same time the requirements for "soft" skills in general are widely diverse. On the one hand, "Support" and "Analytics" specialists have to possess lots of non-technical skills. On the other hand,

**Table 3  Key pairs of "hard" skills by job occupations.**

| Cluster name | Skill #1 | Skill #2 | Jaccard index |
|---|---|---|---|
| Analytics | financial statements | Reporting | 0.45 |
| | 1C | accounting | 0.36 |
| | 1C | 1c programming | 0.35 |
| | accounting | trade management | 0.34 |
| | BPMN | UML | 0.30 |
| Big Data & ML | imap | pop3 | 1.00 |
| | imap | SMTP | 0.84 |
| | pop3 | SMTP | 0.84 |
| | CD | CI | 0.61 |
| | NumPy | Pandas | 0.41 |
| Databases | IBM | Lotus Notes | 0.49 |
| | ASP.NET | C# | 0.41 |
| | .NET | C# | 0.40 |
| | .NET | ASP.NET | 0.36 |
| | ASP.NET | MVC | 0.24 |
| Engineering | instrumentation | process control system | 0.20 |
| | compass 3d | SolidWorks | 0.16 |
| | automation of processes | process control system | 0.14 |
| | Product Development | the launch of new products | 0.14 |
| | AutoCAD | normative-technical documentation | 0.13 |
| Hardware | IP | TCP | 0.94 |
| | BGP | OSPF | 0.57 |
| | setting up the pc | software setting | 0.51 |
| | configuring dns | setting up the pc | 0.46 |
| | server administration | Windows | 0.42 |
| SEO | promotion | website promotion | 0.54 |
| | contextual advertising | Yandex Direct | 0.47 |
| | branding | promotion | 0.44 |
| | SEO | technical website audit | 0.41 |
| | contextual advertising | web analytics | 0.37 |
| Support | 223-fz [procurement law] | 44-fz [public procurement law] | 0.61 |
| | staff training | Training | 0.58 |
| | customer engagement | sales | 0.37 |
| | B2B | customer engagement | 0.35 |
| | B2B | sales | 0.32 |
| Testing | CSS | HTML | 0.74 |
| | HTML | JavaScript | 0.51 |
| | Functional testing | regression testing | 0.47 |
| | MySQL | PHP | 0.46 |
| | CSS | JavaScript | 0.44 |
| WebDev | UI | UX | 0.66 |
| | Illustrator | Photoshop | 0.27 |
| | Graphic Design | Web Design | 0.27 |
| | CorelDRAW | Graphic Design | 0.27 |
| | editorial activities | proofreading | 0.24 |

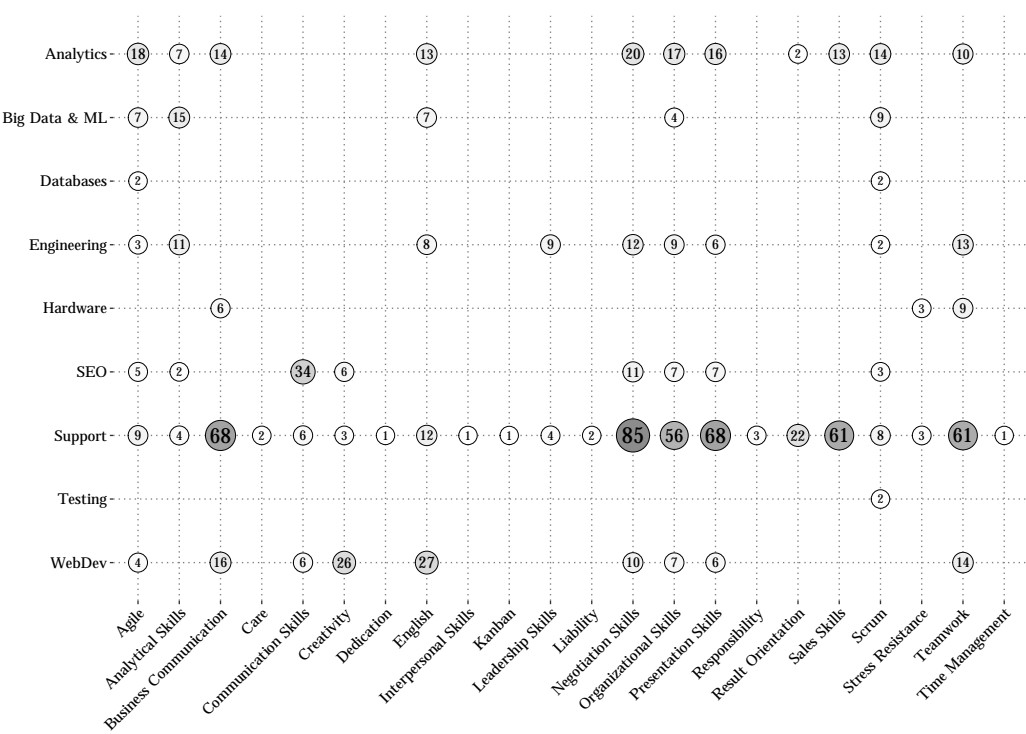

**Figure 4  Average Confidence grid for "soft" skills by job occupations.**

"Testing", "Database", and "Hardware" occupations have no strong requirements for such competencies. Moreover, the dominance is preserved for the majority of the specialties of the "hard" skills. In this case only 20–30% of the required combinations of competencies include non-technical skills.

## DISCUSSION

Analysis of job advertisements data includes stages of skills processing and identification, skill set mapping, and knowledge extraction. In this study, in order to identify the interrelation between "soft" and "hard" skills for IT professionals, the competencies were extracted and standardized in accordance with natural language processing and skill base mapping. Consequently, skill types were separated into two dictionaries of technical and non-technical competencies. Then, "hard" skills were gathered into skill sets related to particular job occupations, using hierarchical cluster analysis. As a result, nine groups of technical skills were identified. Next, each skill cluster was mapped with already processed "soft" skills. Finally, association mining was introduced for the creation of the grid map indicating the linkage between job occupations and non-technical skills.

According to the results, the obtained skill sets (related to job occupations) have disjoint redistribution of processed technical skills. On the one hand, it allows to separate job positions with mixed-tasks and concentrate on the pure relation to particular non-technical skills. On the other hand, this approach does not capture particular job positions

**Table 4   Rules with maximum Confidence by "soft" skills.**

| # | rhs | Cluster name | lhs | Confidence | Lift | N |
|---|---|---|---|---|---|---|
| 1 | Agile | Analytics | PMBOK | 0.43 | 23.64 | 436 |
| 2 | Analytical Skills | Engineering | System Analysis | 0.18 | 8.37 | 288 |
| 3 | Business Communication | Support | business correspondence, contracts, Phone Calls, work with a large amount of information | 0.97 | 17.15 | 167 |
| 4 | Care | Support | work with a large amount of information | 0.05 | 7.33 | 265 |
| 5 | Communication Skills | SEO | advertising campaigns, marketing research, promotion, Strategic Marketing | 0.75 | 21.19 | 148 |
| 6 | Creativity | WebDev | editing, proofreading, writing skills | 0.42 | 36.21 | 155 |
| 7 | Dedication | Support | sales | 0.01 | 1.63 | 173 |
| 8 | English | WebDev | translation | 0.80 | 11.25 | 869 |
| 9 | Interpersonal Skills | Support | Phone Calls | 0.02 | 2.14 | 222 |
| 10 | Kanban | Support | Project Manager | 0.01 | 5.62 | 222 |
| 11 | Leadership Skills | Engineering | assigning tasks to developers | 0.15 | 21.24 | 248 |
| 12 | Liability | Support | customer engagement, sales | 0.02 | 14.98 | 174 |
| 13 | Negotiation Skills | Support | B2B, business correspondence, contracts, customer engagement, Phone Calls, Project Manager, Sales Management | 1.00 | 12.01 | 217 |
| 14 | Organizational Skills | Support | business correspondence, Event Management, personnel management | 0.92 | 21.71 | 157 |
| 15 | Presentation Skills | Support | B2B, contracts, powerpoint, sales | 0.96 | 21.58 | 173 |
| 16 | Responsibility | Support | work with a large amount of information | 0.08 | 2.64 | 435 |
| 17 | Result Orientation | Support | Customer Support, sales, Sales Planning | 0.81 | 43.53 | 148 |
| 18 | Sales Skills | Support | Customer Support, Sales Management, Sales Planning | 0.92 | 30.00 | 150 |
| 19 | Scrum | Analytics | PMBOK | 0.30 | 19.18 | 305 |
| 20 | Stress Resistance | Support | work with a large amount of information | 0.07 | 4.52 | 405 |
| 21 | Teamwork | Support | business correspondence, Phone Calls, Project Manager, work with a large amount of information | 0.95 | 11.36 | 172 |
| 22 | Time Management | Support | business correspondence | 0.01 | 1.77 | 171 |

but focuses on the professional area of competence. The last-mentioned allows identifying in-demand skills on the broader level of investigation (*Pejic-Bach et al., 2020*). For instance, in this study, the most demanded "hard" skills relate to technologies used in web service development and testing (HTML, JavaScript, PHP, CSS). Namely, the job occupation including mentioned competencies I call "Testing". Related work gets approximately the same distribution of frequent "hard" skills on cross-country level. However, the target cluster is named "Web Technicians", "Web Programming" or "Software Development" (*Giabelli et al., 2020*; *Jia et al., 2018*; *Hiranrat & Harncharnchai, 2018*; *Wu, Shi & Yang, 2017*; *Papoutsoglou, Mittas & Angelis, 2017*; *Ternikov & Aleksandrova, 2020*). The main distinctions are based on the aggregation level of job advertisements and the usage of mixed skill sets. The provided approach relates to the sub-division of skills needed for software development. Hence, it focuses more on skill-cluster rather than on specific job position names.

Nevertheless, the most required technical skills and their combinations in the IT labor market vary among different regions and job postings platforms. Existing studies mention that competencies are regionally specific (*Hiranrat & Harncharnchai, 2018*; *Giabelli et al., 2020*). For example, in the CIS region such skills are "1C" (ERP system), "compass 3d" (3D modeling software), and "Yandex Direct" (platform for context advertising). Some of them are highlighted in the studies that analyze the HeadHunter database (*Botov et al., 2019*; *Ternikov & Aleksandrova, 2020*). In addition, all frequent skills mentioned in prior studies are presented in this research. Note that the frequency of occurrence is slightly diverse depending on the level of job postings aggregation.

According to the findings, the most in-demand non-technical skills are the following: "Teamwork", "Negotiation Skills", "Business Communication", and "English". Taking into account the most accurate combinations of competencies related to occupational skill sets, I can also outline "Agile", "Scrum", "Organizational Skills", "Presentation Skills", and "Analytical Skills". Generally, these results do not contradict the existing research (*Chaibate et al., 2019*; *Matturro, 2013*; *Steinmann, Voigt & Schaeffler, 2013*; *Daneva, Wang & Hoener, 2017*; *Giabelli et al., 2020*; *Hiranrat & Harncharnchai, 2018*; *Skhvediani et al., 2021*; *Papoutsoglou, Mittas & Angelis, 2017*; *Wu, Shi & Yang, 2017*; *Gurcan & Cagiltay, 2019*; *Ahmed, Capretz & Campbell, 2012*). Actually, I can again pose the regional specificity of in-demand non-technical skills. Interestingly, in contrast to the findings, "Communication Skills" are topped in the majority of studies, but "Teamwork" is less required. However, due to slightly different formulations provided in several job ads databases, I can conclude that some names of skills could be aggregated, *e.g.*, "Communication Skills" and "Business Communication"; "Leadership" and "Organizational Skills", "Interpersonal Skills" and "Teamwork".

## Threats to validity

The study faces some limitations that should be acknowledged. Accordingly, I overview the possible validity threats in more detail. Namely, internal, external, and construct validity.

### *Internal validity*

In the study, I overcome several limitations which stated in related research: narrow data period, small sample size, use of mono-language ads, selection bias (*Pejic-Bach et al., 2020*; *Brooks, Greer & Morris, 2018*; *Skhvediani et al., 2021*; *Papoutsoglou, Mittas & Angelis, 2017*; *Cao et al., 2021*). However, data structure limitations should be considered. The data are comprised of platform-specific job postings. On the one hand, the vacancies are obtained from the homogeneous source of data. On the other hand, the job platform allows recruiters not to fill in specific skills field that is used for the analysis. Accordingly, some job postings contain only basic skills or broad skill sets. In addition, the demand for skills varies for different regions representing the IT vacancies. Moreover, there is no common principle to assign specific professional areas to a particular vacancy. Thus, HR specialists are not fully able to precisely specify the occupation using the HeadHunter professional area classifier.

*External validity*

Obtained results are generalizable beyond the experimental setting. However, the demand for IT specialists in the labor market is not restricted by only one hiring platform. Additional data sources should be used in further investigations. Moreover, the internal recruiting processes inside many companies are not publicly disclosed. Accordingly, companies may promote themselves and post false recruitment information. In future research, the hidden employment might be taken into account using company survey data enrichment.

*Construct validity*

The research design is based on data-driven textual processing of skills and vacancies. The noise in data, including different notations of the same skills, was partly reduced by the skill standardization procedure. Limitations that could be highlighted for the clustering approach include an empirically chosen number of clusters, the need for additional manual processing of extracted words and phrases, and a strong connection between the quality of available data and expected results. Overall, relatively low relation of "soft" skills to more computationally intensive "hard" skills could be the consequence of vacancy postings structure and the dominance of technical competencies there. Related research also proposes the effect of superiority of technical skills over non-technical (*Skhvediani et al., 2021*; *Wu, Shi & Yang, 2017*; *Papoutsoglou, Mittas & Angelis, 2017*). Thorough data processing is needed to perform reasonable competencies allocation in certain job positions in case of skills detection (*Daneva, Wang & Hoener, 2017*; *Pejic-Bach et al., 2020*). The results indicate that skills from one cluster may relate to different purposes, *e.g.*, knowledge of specific protocols, programming languages with their libraries, special software, integrated environments, frameworks, managerial or analytical tools. In this study, the difference between skills' purposes is not analyzed but it could be added for the further implications of the applied methodology.

# CONCLUSION

This research gives a broader understanding of the demand-side requirements on the CIS labor market in the IT sphere. The most accurate combinations of "soft" and "hard" skills are obtained and ready for implementation in educational curriculum and professional standards. Furthermore, the applied methodology allows transferring the data analysis techniques into other regional markets. Thus, the highlighted methodology has a high potential to be used both in a theoretical and practical manner.

Summing up, the main practical implications relate to the modification of HR and educational policies. I provide a novel approach for skills grid visualization that may help to identify the most in-demand skill sets. Moreover, the provided methodology helps to gather information for in-time labor market monitoring. The last undoubtedly can foster the curricula development and modernization to ensure the proper hiring.

This paper provides an approach for key skills detection in different job occupations that is scalable for similar data structures of the labor market. The novelty of the research design relates to the pipeline, namely, steps of unstructured text standardization, key competencies extraction, application of hierarchical clustering based on dependencies between different

skills, and the use of association mining techniques for obtaining knowledge on job requirements.

Based on the presented analysis, further studies could explore the following directions. Firstly, research on the dynamics of skill sets changes. Secondly, an overview of the other sectors of the labor market. Thirdly, classification of job occupations and skills with the use of already existing state classifiers.

### Funding
The authors received no funding for this work.

### Competing Interests
The authors declare there are no competing interests.

### Author Contributions
- Andrei Ternikov conceived and designed the experiments, performed the experiments, analyzed the data, performed the computation work, prepared figures and/or tables, authored or reviewed drafts of the paper, and approved the final draft.

### Data Availability
The raw data with job ads are available at figshare:

Ternikov, Andrei (2022): it_vacancy_data. figshare. Dataset. https://doi.org/10.6084/m9.figshare.19005092.v1

The processed sample with job ads, skills, and similarities are available at figshare:

Ternikov, Andrei (2021): vacancy_skills_data. figshare. Dataset. https://doi.org/10.6084/m9.figshare.17075717.v1.

### Supplemental Information
Supplemental information for this article can be found online at http://dx.doi.org/10.7717/peerj-cs.946#supplemental-information.

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
