# Peer review of "Soft and hard skills identification: insights from IT job advertisements in the CIS region"

_PeerJ Computer Science, doi:10.7717/peerj-cs.946_

## Round 0.1 · original submission · Major Revisions

Both reviewers agreed that the paper is interesting and well-written in general. Both make important comments on how to improve the paper. Therefore, I would like you to go through these and incorporate them as well as possible. For me, the most critical issues are:
- Make context clear early on
- Discuss threats to validity explicitly (probably in separate section)
- Make distinction to previous paper explicitly clear
- Make also the data available, or argue why this is not possible.

Finally, I suggest you rethink the names "hard" and "soft". I think they do not denote the concepts very well. Why don't you go for "technical" and "non-technical" as you use that already to describe your terms?

Reviewer 1 ·

Basic reporting

- In general, I found the manuscript to be mostly well written. The text could however benefit from some academic proofreading with regards to more minor stuff related to punctuation and minor readability edits.

-While Table 1 is briefly introduced at the start of “Related Work”, it could be introduced better to the reader. Currently, you mention many of these terms in the starting paragraph, but how they are shown in Table 1 is a bit obscure at first. In my opinion, by having a paragraph strictly introducing the structure of Table 1, the readability of the paper would be improved. The way the table classifies prior work based on sample size and used approach is commendable, however.

-All other Tables could also be introduced to the reader in a more comprehensive manner.

- How did you get the cluster names for Tables 2, 3, and 4? Did you name the clusters by yourself? This could be mentioned briefly in the manuscript.

Experimental design

- Three research goals are mentioned in the introduction section, and the author has results for these goals in the results section.

- How does this paper differ from the paper “Demand for skills on the labor market in the IT sector”, by the same author? By glancing over the previous paper, you also cite in your manuscript, it seems the novelty lies in the hard and soft skill classification and the association rule mining. I think the paper would be improved by clearly stating the differences between these two papers, perhaps towards the end of the introduction, near your contributions, with a sentence of two.

- I find the way you report soft and hard skills identification confusing. How does semi-manual identification work exactly? Do you use the same list as Calanca et al.? If so, will you miss soft skills not contained in the list? Is this a validity threat that should be discussed?

- While the code used to analyze the data is shared, the data itself is not. Thus, I was not able to rerun the analysis done by the author. Sharing the data with the completed manuscript would greatly improve the reproducibility of the work.

Validity of the findings

- Personally, I didn't find anything surprising in your results. You compare your results to previous findings, noting the region specificity in certain skills. However, I don't see if your results miss something from previous results. That is, are some skills which are present and popular in previous findings not present in your results? Investigating this could perhaps expand both the results and discussion sections.

- You mention that the study was done on the data related to the Commonwealth of Independent States. In my opinion, this contextual information is important enough, so it could be added to the title, e.g., “Insights from IT Job Advertisements in the CIS region”. At the very least, I think this should be mentioned in the abstract.

- While the code used to analyze the data is shared, the data itself is not. Thus, I was not able to rerun the analysis done by the author. Sharing the data with the completed manuscript would greatly improve the reproducibility of the work.

- Some limitations on the discussion section are already mentioned, but some of them are very brief. This could be expanded to its own section, e.g. internal, external, and construct validity.

Reviewer 2 ·

Basic reporting

Summary: The author of this paper studied the demand for skills in the IT-sphere to discover the mapping between required skill-sets and job occupations.
The proposed methodology for skills identification uses natural language processing, hierarchical clustering, and association mining techniques.
The dataset used covers the 2015–2019 period, with 351,623 observations. A set of 3,034 unique frequent skills was extracted and prepared for further standardization. 

The results show explicit information about the combinations of "soft" and "hard" skills required for different professional groups.
The author explains the goal of the study highlighting that the findings provide valuable insights for supporting educational organizations, human resource (HR) specialists, and state labor authorities in the renewal of existing knowledge about skill-sets for IT professionals.


The paper is well structured and easy to follow. The related works are well presented, putting the study in context. The raw data have been shared, along with the code used to perform the study. The results include the definitions of all terms.

Experimental design

The research is within the aims and scope of the journal.

The research questions are defined on page 2. To help the reader the author might restructure the goals of the paper in a question format, assigning the number to each question, and providing a specific answer to each question in the results section.

The goals of the study are relevant and meaningful, and it is stated how this study fills the gap in the literature. The methodology is well explained and performed to a high technical standard, and all the methods are described with sufficient details which allow replication studies.

Validity of the findings

All the underlying data have been provided, and the statistics explained.
However, I strongly encourage the author to dedicate a specific section of the paper to better explain the limitations of the study (threats to validity).
Conclusions are well stated and linked to the research questions, but the author could help the reader in re-organizing the RQ as suggested in the previous section.

Additional comments

Overall, the paper is well written and interesting. It fits the journal, but the presentation can be improved.

---

## Round 0.2 · accepted · Accept

Thank you very much for the revision of the paper. The reviewers and I agree that you incorporated all feedback that we gave. We are happy to recommend accepting the paper!

Reviewer 1 ·

Basic reporting

The author has improved on all points raised in my previous review well. Most commendable are the new threats section and the shared raw data. In my opinion, the manuscript can be published.

Experimental design

-

Validity of the findings

-

Reviewer 2 ·

Basic reporting

The paper is clear and well written.

Experimental design

The research is original and within Aims and Scope of the journal.

Validity of the findings

All the data have been provided and the analysis are well explained.

Additional comments

Thanks to the author for the revised version of the paper. I feel that the comments have been addressed and I think the paper can be accepted.